# Causal Clustering for Conditional Average Treatment Effects Estimation and Subgroup Discovery

Zilong Wang
*Industrial and Systems Engineering*
*Georgia Institute of Technology*
Atlanta, Georgia, USA
* zwang937@gatech.edu

Turgay Ayer
*Industrial and Systems Engineering*
*Georgia Institute of Technology*
Atlanta, Georgia, USA
† ayer@isye.gatech.edu,

Shihao Yang
*Industrial and Systems Engineering*
*Georgia Institute of Technology*
Atlanta, Georgia, USA
‡ shihao.yang@isye.gatech.edu

*Abstract*—**Estimating heterogeneous treatment effects is critical in domains such as personalized medicine, resource allocation, and policy evaluation. A central challenge lies in identifying subpopulations that respond differently to interventions, thereby enabling more targeted and effective decision-making. While clustering methods are well-studied in unsupervised learning, their integration with causal inference remains limited. We propose a novel framework that clusters individuals based on estimated treatment effects using a learned kernel derived from causal forests, revealing latent subgroup structures. Our approach consists of two main steps. First, we estimate debiased Conditional Average Treatment Effects (CATEs) using orthogonalized learners via the Robinson decomposition, yielding a kernel matrix that encodes sample-level similarities in treatment responsiveness. Second, we apply kernelized clustering to this matrix to uncover distinct, treatment-sensitive subpopulations and compute cluster-level average CATEs. We present this kernelized clustering step as a form of regularization within the residual-on-residual regression framework. Through extensive experiments on semi-synthetic and real-world datasets, supported by ablation studies and exploratory analyses, we demonstrate the effectiveness of our method in capturing meaningful treatment effect heterogeneity.**

*Index Terms*—**clustering, heterogeneous treatment effect, causal inference, kernel, subgroup discovery**

## I. INTRODUCTION AND MOTIVATION

### A. Heterogeneity in Treatment Effects

Causal inference aims to determine the effect of a specific intervention or treatment on an outcome of interest [1],[2]. This is particularly valuable in fields such as healthcare, economics, and social sciences, where accurately estimating the effects of policies or interventions is essential. Using the potential outcomes framework [2], we observe a dataset of $n$ independent and identically distributed samples $\{(X_i, Y_i, W_i)\}_{i=1}^n$, where $X_i \in \mathcal{X} \subseteq \mathbb{R}^p$ represents individual features, $Y_i \in \mathbb{R}$ is the observed outcome, and $W_i \in \{0, 1\}$ denotes the treatment assignment. For this binary treatment, the average treatment effect (ATE) is defined as $\mathbb{E}[Y^{(1)} - Y^{(0)}]$, where $Y^{(w)}$ represents the potential outcome that would have been observed under treatment $W = w$. However, treatment effects often vary across subgroups in both magnitude and direction. A treatment beneficial for one subgroup may be harmful for another, and reporting only the ATE can obscure such heterogeneity [3].

To address this, a common target of interest is the conditional average treatment effect (CATE), defined as

$$\tau(x) := \mathbb{E}[Y^{(1)} - Y^{(0)} | X = x],$$

which enables personalized treatment effect estimation.

A wide range of methods has been developed to estimate CATE, including recursive partitioning [4],[5],[6], partially linear models [7], and neural networks [8], among others. Additionally, meta-learners, such as the X-Learner [9] and the R-Learner [10], integrate multiple approaches to enhance estimation accuracy.

### B. Motivation and Current Work

Understanding how different patients respond to treatment is a central challenge in biomedical research, especially in observational studies where treatment assignment is not randomized. In such settings, investigators often seek to identify patient subgroups that exhibit heterogeneous treatment effects to first inform clinical decision-making **before** deciding on personalized intervention strategies.

This task is fundamentally exploratory and descriptive: rather than predicting outcomes directly or to be used directly in treatment assignment optimization, it is meant to be an addition to the toolkit for providing interpretable and high-level insights. One approach is to analyze the Conditional Average Treatment Effect (CATE) as a function of baseline covariates and uncover structure in its variation. In this work, we explore how unsupervised clustering applied to estimated CATEs can facilitate such subgroup discovery.

Clustering offers a flexible, interpretable, and data-driven way to visualize and summarize treatment heterogeneity. Unlike methods that threshold scalar CATE values (e.g., treating only patients with $\hat{\tau}(X) > 0$), clustering does not require arbitrary cutoffs or monotonicity assumptions. This is particularly valuable in new diseases or heterogeneous populations where no prior guidance exists. Clustering on regularized or kernel-smoothed CATE estimates helps mitigate estimation noise and

reveals subgroups with similar treatment responses—unlike standard feature-based clustering, which may ignore treatment effect heterogeneity. This assists practitioners in identifying clinically relevant subgroups for further targeted investigations or data collection.

While clustering is a well-established tool in unsupervised learning, its application to causal inference remains relatively nascent. The key challenge stems from the **fundamental problem of causal inference** [2]: we never observe both treated and untreated outcomes for the same individual, necessitating an estimation step before any subgrouping can be applied. This leads to a two-stage procedure: first estimating individual treatment effects, then analyzing their structure to uncover latent subpopulations with heterogeneous responses.

Several strands of related work have pursued this two-stage strategy. One direction focuses on clustering with latent or noisy data, such as Gaussian mixture models trained via the Expectation-Maximization (EM) algorithm [11],[12],[13],[14], and model-based clustering under measurement error [15],[16]. These lines of work, while not restricted to causal inference in application, do not handle adjusting estimates when propensity is unbalanced. More recently, [17] proposed a joint EM framework that clusters based on radial basis function (RBF) distances between CATE estimates, with the distance weights learned via a transformer architecture. Another direction involves explicitly estimating the CATE and then assigning subgroups through decision rules or clustering. For example, [18],[19],[20] perform recursive partitioning on the estimated CATE surface to derive interpretable subgroup assignments. For the tree based rules such as `CRE` by [19], there is no direct way to set the number of desired clusters. In our work, we believe in allowing users to flexibly incorporate a variety of existing libraries and solvers at each step.

Interestingly, while tree-based methods such as classification and regression trees, random forests [21], and causal forests [5] are widely used for both CATE estimation and subgroup discovery, often by leveraging splitting rules, there has been limited exploration of using these ensembles as explicit kernel density estimators for subgroup discovery, which presents an opportunity to incorporate kernel learning techniques. Forest-based methods are well known to induce similarity metrics that behave like adaptive local kernel estimators [22],[23],[5],[6]. In the traditional unsupervised learning literature, these forest-induced kernels have even been used in spectral clustering algorithms [24], suggesting potential synergy in the causal setting that motivates this research.

### C. Our Contribution: A New Perspective

Therefore, in this work, we introduce a novel perspective on subgroup discovery for heterogeneous treatment effects that informs the clustering process using a learned kernel (similarity matrix) derived from residual-on-residual (R-Learner) style estimators [10]. We frame the hard clustering step as a form of implicit regularization on the CATE estimation task, promoting smoothness and subgroup coherence in the estimated treatment effects. We view this work as a first step toward a broader class of tools that combine kernel learning and causal effect estimation in a modular and interpretable way. We believe this method represents a promising direction for exploratory analysis of treatment effect heterogeneity.

Our primary contributions are threefold. First, we propose a kernel-based approach for identifying subpopulations with distinct treatment responses, offering a new lens for exploratory subgroup discovery in CATE analysis. Second, we introduce a practical and modular framework that can be implemented using standard machine learning libraries and workflows, allowing for straightforward integration with existing model selection and tuning pipelines. Finally, we provide empirical support through an ablation study on a semi-synthetic dataset, investigate the performance of common cluster size selection heuristics for our setup, evaluate the potential performance loss relative to the original CATE estimators on a fully synthetic benchmark, and demonstrate the broader applicability of our method beyond randomized control trials on an observational dataset of emergency health records with treatment assignment confounding.

### D. Organization

The remainder of this paper is organized as follows. In §II, we provide our setup, a background on the R-learner framework, and a brief review of clustering. Next, in §III, we present our proposed framework. Then in §IV describes the experimental setup and results, demonstrating the effectiveness of our approach on synthetic and real-world datasets. Finally, §V concludes with a discussion of our findings and implications.

## II. SETUP AND PRELIMINARIES

### A. The Robinson Decomposition of the CATE

In order to identify the CATE $\tau(x)$, we rely on the following commonly used identification strategies in [25]:

**A1 (Consistency)**: The observed outcome for an individual under treatment $W_i$ corresponds to the potential outcome under that treatment level, i.e., $Y_i = Y_i^{(1)} \cdot \mathbb{I}(W_i = 1) + Y_i^{(0)} \cdot \mathbb{I}(W_i = 0)$.

**A2 (Unconfoundedness)**: Also known as the ignorability assumption, this assumes that, conditional on observed covariates, the potential outcomes are independent of the treatment assignment, i.e., $\{Y_i^{(1)}, Y_i^{(0)} \perp W_i \mid X_i\}$.

**A3 (Positivity)**: We define the propensity score as the conditional probability of receiving treatment given covariates $x$: $e(x) = P(W = 1 \mid X = x)$. The positivity assumption requires that: $0 < e(x) < 1, \quad \forall x$, i.e., all individuals have a strictly non-zero probability of being in the treatment or control groups.

We now review the Robinson decompostion by [26]. To aid readability, we rewrite the following terms below as:

$$\mu_{(w)}(x) := \mathbb{E}[Y^{(w)}|X=x], \forall w \in \{0,1\} \tag{1}$$

$$\varepsilon_i(w) := Y_i^{(w)} - \{\mu_{(0)}(X_i) + w\tau(X_i)\} \tag{2}$$

$$m(x,w) := \mathbb{E}[Y|X=x, W=w] \tag{3}$$

$$= \mu_{(0)}(x) + e(x)\tau(x) + \underbrace{\mathbb{E}[\varepsilon(w)|X=x]}_{=0 \text{ by unconfoundedness}} \tag{4}$$

$$= \mu_{(0)}(x) + e(x)\tau(x) \tag{5}$$

We then obtain the residualized form in Equation (6):

$$\underbrace{Y_i - m(X_i, W_i)}_{\text{Outcome Residual } (\tilde{Y}_i)} = \underbrace{\{W_i - e(X_i)\}}_{\text{Propensity Residual } (\tilde{W}_i)} * \underbrace{\tau(X_i)}_{CATE} + \varepsilon_i(W_i), \tag{6}$$

This decomposition, originally used by [26] for partially linear models, has seen a considerable revival of interest in recent years and it has been used in many popular and seminal works such as the Double Machine Learning (DML) framework [27] and forest ensemble methods such as the Orthogonal Random Forest (ORF) [28], Causal Forests (CF) [5]. In particular, 6 can be re-formulated as a regularized empirical loss minimization program below in Equation (7):

$$\hat{\tau}(\cdot) := \underset{\tau \in \mathcal{F}}{\operatorname{argmin}} \left( \frac{1}{n} \sum_{i=1}^{n} \{\tilde{Y}_i - \tilde{W}_i \tau(X_i)\}^2 + \Lambda(\tau(\cdot)) \right). \tag{7}$$

Here $\Lambda$ is a regularization term that penalizes the complexity of the functional form of the CATE decision variable $\tau$ either explicitly (e.g., penalized regression) or implicitly (e.g., complexity of a neural network, tree depths of a forest). This residual-on-residual regression approach has incredible flexibility in allowing the propensity $e(\cdot)$ and mean outcomes $m(\cdot)$ to be estimated separately with a variety of supervised machine learning techniques. In addition, it has very attractive statistical properties when the estimation is done with proper sample-splitting and cross-fitting techniques (cf. DML by [27], Neyman Orthogonalization by [29]). For our work, we are interested in building upon the R-Learner framework introduced by [10] through the causal forest first introduced by [4]. The former specifically studies the properties for Equation (7) when $\tau$ belongs to the Reproducing Kernel Hilbert Space (RKHS) induced by a positive semi-definite kernel function $K$ and thus admits a large variety of underlying estimators such as linear models, kernel ridge regression, and even more complex non/semi-parametric models such as ensembles. Tree ensemble methods frame the estimation problem as a locally weighted regression task, where the weights are implicitly defined by the frequency with which a sample falls into the same leaf as other samples (referred to as forest weights). These forest ensemble methods minimize the residualized loss around an input sample with features $x$ as follows:

$$\hat{\tau}(x) = \underset{\tau}{\operatorname{argmin}} \sum_{j=1}^{n} \alpha_j(x) \left\{ Y_j - \hat{m}_x - (W_j - \hat{e}_x) \cdot \tau(x) \right\}^2 + \Lambda(\tau(x)),$$

where $\alpha_j(x)$ represents the forest weights matrix (which will be formally defined in Section III). Notably, both the response function $\hat{m}_x$ and the propensity function $\hat{e}_x$ are locally estimated, as indicated by the subscripts with $x$, and debiased by means of cross-fitting [27] (also called "honest-fitting"). The regularization penalty term $\Lambda(\tau(x))$ controls the complexity of the treatment effect function $\tau(\cdot)$, and it is implicitly determined by various tree parameters such as maximum depth, minimum leaf size, and tree balance. Crucially, this framework enables the learning of a kernel that captures the similarity between sample points based on their Conditional Average Treatment Effect (CATE). This kernel provides an intuitive mapping of the relationship between samples, which is instrumental in the subsequent step of kernelized clustering.

### B. Clustered Orthogonal Learner

Given a hard clustering assignment of $k$ clusters: $C = \{C_1, C_2, \ldots, C_k\}$ over the indices of our $n$ samples $[n] = \{1, 2, \ldots, n\}$, said cluster assignment must form a **partition** over the set of sample indices $[n]$ i.e.,:

1) **Disjoint Clusters:** $C_i \cap C_j = \emptyset, \forall i \neq j \in [k]$
2) **Complete Coverage:** $\bigcup_{j=1}^{k} C_j = [n]$
3) **Non-empty Clusters:** $\forall j \in [k] : C_j \neq \emptyset$

While the above description is typically more intuitive, it is more convenient and insightful to formulate our clustering problem via the equivalent Sum-of-Norms (SON) model [30] as follows:

$$\hat{\tau}(X_i) = \underset{\tau \in \mathcal{F}}{\operatorname{argmin}} \left( \frac{1}{n} \sum_{i=1}^{n} \{\tilde{Y}_i - \tilde{W}_i \tau(X_i)\}^2 \right. \tag{8}$$

$$\left. + \lambda \sum_{i,j \in [n]: i < j} \|\tau(X_i) - \tau(X_j)\|_q \right), \tag{9}$$

where the penalty term, controlled by $\lambda \geq 0$, enforces similarity between treatment effect estimates across samples. The choice of $q = 0$ (pseudo-norm) explicitly limits the number of unique centroids, effectively determining the number of clusters. When $\lambda = 0$, the model reduces to the original un-clustered orthogonal learner, while increasing $\lambda$ encourages greater clustering, reducing the number of distinct centroids (with $\lambda \to \infty$ forcing the estimator $\hat{\tau}$ to output the same estimate for every $X_i$). This reformulation also aligns itself with the residual-on-residual structure of the R-Loss in Equation (7), with the number of clusters i.e., total number of unique CATE estimates forming a natural regularizing component. Nevertheless, it presents a computational and theoretical challenge as optimal k-means clustering—an instance of the Minimum Sum-of-Squares Clustering (MSSC) problem—is known to be NP-hard [31]. To address this, we introduce a relaxation of Equation (9) along with a practical implementation in the following section.

### III. PROPOSED PROCEDURE

To address the computational challenges of the hard clustering formulation in Equation (9), we introduce a two-step

relaxation that decouples the optimization problem into sequential stages:

**Data Splitting** We begin by partitioning the indices of the dataset of $n$ samples into $S$ mutually exclusive folds $\mathcal{I}_1, \ldots, \mathcal{I}_S$. For each fold $s$, let $\mathcal{I}_{-s}$ denote the complement set (i.e., index of all samples not in fold $\mathcal{I}_s$).

**Step 1: Estimation of CATE and Kernel**

For each fold $s = 1, \ldots, S$, we solve the residual-on-residual loss on the training subset $\mathcal{I}_{-s}$ to obtain out-of-fold treatment effect estimates and learned kernels:

$$\hat{\tau}^{(-s)}(\cdot) := \underset{\tau \in \mathcal{H}}{\arg\min} \left( \frac{1}{|\mathcal{I}_{-s}|} \sum_{i \in \mathcal{I}_{-s}} \{\tilde{Y}_i - \tilde{W}_i \tau(X_i)\}^2 + \Lambda(\tau) \right),$$
$$\tag{10}$$

and construct out-of-fold predictions:

$$\hat{\tau}_i^{(\text{HF})} := \hat{\tau}^{(-s)}(X_i), \quad \text{for } i \in \mathcal{I}_s. \tag{11}$$

Similarly $\forall i \in \mathcal{I}_{s_1}, j \in \mathcal{I}_{s_2}$, we compute the learned kernel, evaluated in cross-fitted fashion by extracting them from the trained out-of-fold estimators:

$$\hat{K}^{(\text{HF})}(X_i, X_j) := \frac{1}{2}\left( \hat{K}^{(-s_1)}(X_i, X_j) + \hat{K}^{(-s_2)}(X_i, X_j) \right) \tag{12}$$

The complexity of the learned estimator is implicitly regularized and tuned (e.g., tree-depth, imbalance, minimal leaf size for forest based models).

**Step 2: Clustering Using Cross-Fitted Estimates**

We now apply kernelized convex clustering using the cross-fitted treatment effect estimates $\hat{\tau}_i^{(\text{HF})}$ and similarity kernel $\hat{K}^{(\text{HF})}$. We solve:

$$\min_{U \in \mathbb{R}^n} \sum_{i=1}^n \|U_i - \hat{\tau}_i^{(\text{HF})}\|_2^2 + \lambda \sum_{i<j} \hat{K}^{(\text{HF})}(X_i, X_j)\|U_i - U_j\|_1. \tag{13}$$

to obtain the final estimates $U_i$ and hard cluster membership assignments ($\forall i, j \in [n] : U_i = U_j \iff i, j$ in same cluster).

Computationally, this sequential approach relaxes the original problem in two key ways: First it transforms the simultaneous optimization of treatment effect estimation and clustering into a more tractable sequential process. Second, the NP-hard, non-convex clustering problem is replaced with clustering methods that can be solved using traditional iterative procedures e.g., Lloyd's algorithm [32] or as a semi-definite program via methods by [33],[34],[35].

By structuring the optimization in this manner, we balance computational feasibility with the flexibility of kernel-based clustering while preserving the algorithmic flexibility of the initial estimation step, making implementation with out-of-the box libraries straightforward. More importantly, by decoupling the initial learning step and subsequent clustering step and allowing for honest estimation, the clustering analysis (Step 2) can also be conducted on a new set of unseen test data points without retraining the estimators of (Step 1), resulting

in reduced inference times. This modification also allows us to conduct honest estimation [27],[10],[5],[29], where separate data subsets are used in cluster construction and estimation, thus avoiding overfitting and preserving the integrity of subgroup discovery. We now describe how to obtain a valid learned kernel from the trained $\tau(\cdot)$ estimator.

### A. Forest Based Kernels

To effectively utilize the capabilities of tree-based ensemble methods, we leverage the `grf` package in `R` [5]. Specifically, we can construct a valid kernel matrix by extracting and transforming the "forest weights matrix" from a trained forest model. Given a forest of $B$ trees trained on $n$ data points and $m$ new test data points, which may include previously unseen instances, we derive a non-negative $m \times n$ similarity matrix $\alpha$ that quantifies the relationship between each new test data point with each of the training data points. This relationship is mathematically expressed as follows, adhering to the notation established by [5] and [6] in the following form:

$$\alpha_{j,i} = \alpha_j(X_i) := \frac{1}{B} \sum_{b=1}^{B} \frac{1\{X_j \in L_b(X_i)\}}{|L_b(X_i)|}. \tag{14}$$

Here, the $(j, i)$-th entry of the forest weights matrix $\alpha$, denoted $\alpha_j(X_i)$, serves as a measure of similarity from a training data point $X_i$ to a test data point $X_j$. For each tree $b$, this measure sums the instances where $X_j$ falls into the leaf node containing $X_i$ (denoted as $1\{X_j \in L_b(X_i)\}$) and divides by the total number of leaves that contain $X_i$ (denoted as $|L_b(X_i)|$). To obtain a valid kernel matrix for any new set of $m$ test data points from a forest trained on $n$ data points, we construct the outer-product matrix, resulting in a $m \times m$ kernel matrix:

$$K := \alpha \alpha^T, \tag{15}$$

$$K_{j,k} = \sum_{i=1}^{n} \alpha_j(X_i)\alpha_k(X_i). \tag{16}$$

Intuitively, each entry $K_{j,k}$ in the kernel matrix derived from the forest similarity $\alpha$ quantifies the similarity between test points $X_j$ and $X_k$ based on their relationships with all training data points $X_i$. Specifically, $K_{j,k}$ aggregates the pairwise similarities, where $\alpha_j(X_i)$ and $\alpha_k(X_i)$ denote the similarity of training point $X_i$ to test points $X_j$ and $X_k$, respectively. A high value of $K_{j,k}$ indicates strong evidence from the training samples that the test points are similar or related, suggesting they may exhibit comparable outcomes.

### IV. EXPERIMENTS AND RESULTS

### A. Methods

We evaluate the performance of our proposed clustering methodology on both synthetic and real-world datasets. First, in §IV-B, we conduct an ablation study using a semi-synthetic variant of the widely studied Infant Health and Development Program (IHDP) dataset [36], assessing how different kernel constructions and cross-fitting strategies affect CATE estimation and subgroup quality. Next, in §IV-C, we tackle the problem of cluster size selection by evaluating the performance

of various cluster size selection methods on a synthetic dataset. Following that, §IV-D investigates a synthetic adversarial design in which the data is deliberately constructed without any true underlying clusters, to investigate performance under model misspecification with complex response surfaces and high noise. Finally, in §IV-E, we apply our method to the Synthea [37],[38] dataset involving emergency health records with confounded treatment assignments. This is to demonstrate its ability to provide relevant insights in real-world scenarios beyond randomized controlled trials.

### B. Semi-synthetic Ablation Study

| $k$ | Method | PEHE $\downarrow$ | $V_{\text{within}}$ $\downarrow$ | $V_{\text{out}}$ $\uparrow$ |
|---|---|---|---|---|
| 2 | Cross Fitted | **3.60** | 9.53 | 48.16 |
| | Thresholded | 4.62 | 35.99 | **84.23** |
| | RBF Cross Fitted | 4.18 | 16.62 | 38.01 |
| | RBF Oracle | 4.60 | 84.23 | 21.72 |
| | CRE | 4.05 | **2.04** | 67.59 |
| 3 | Cross Fitted | **3.28** | 4.85 | 121.38 |
| | Thresholded | 4.14 | 18.56 | 50.42 |
| | RBF Cross Fitted | 4.23 | 19.65 | 45.62 |
| | RBF Oracle | 4.10 | 60.94 | 136.05 |
| | CRE | 4.11 | 5.30 | 42.98 |
| 4 | Cross Fitted | **3.19** | **3.15** | 205.81 |
| | Thresholded | 4.13 | 18.48 | 112.95 |
| | RBF Cross Fitted | 4.26 | 20.79 | 49.64 |
| | RBF Oracle | 4.10 | 60.86 | 213.51 |
| | CRE | 4.05 | 16.51 | 123.01 |
| 5 | Cross Fitted | **3.08** | 2.07 | **266.45** |
| | Thresholded | 4.13 | 20.27 | 103.82 |
| | RBF Cross Fitted | 4.20 | 21.19 | 55.39 |
| | RBF Oracle | 4.03 | 59.89 | 228.25 |
| | CRE | 3.91 | **0.19** | 61.88 |
| 6 | Cross Fitted | **3.08** | 1.89 | 268.94 |
| | Thresholded | 4.13 | 20.27 | 185.37 |
| | RBF Cross Fitted | 4.18 | 21.11 | 61.62 |
| | RBF Oracle | 4.04 | 59.96 | 311.01 |
| | CRE | 4.15 | 2.92 | 14.4 |

TABLE I: Ablation study of our proposed method on the semi-synthetic IHDP data. we evaluate the quality of cluster estimates of the CATE and the subgrouping quality of the clusters across $k = 2, \ldots, 6$ clusters. $\uparrow$ and $\downarrow$ indicates if it is better to score higher or lower on that metric respectively.

Our ablation study aims to evaluate the impact of kernel construction and estimation strategies on the quality of CATE-based clustering. The IHDP dataset originates from a randomized clinical trial aimed at improving the cognitive and health outcomes of low-birth-weight, premature infants through intensive child care and home visits. The dataset includes 747 individuals, of whom 139 received the treatment and 608 did not, each characterized by 25 covariates capturing demographic, clinical, and socio-economic factors. The version we use is a semi-synthetic dataset constructed by retaining the original covariates from the study while simulating potential outcomes according to predefined response surfaces and is widely used in causal inference research (cf. [39],[20],[40, 41]).

Our default method is `Cross Fitted`, which uses out-of-fold predictions to construct the kernel matrix from learned CATE weights, mitigating overfitting and improving generalization. We compare this against several ablations. First, the

`Thresholded` method mimics the "thresholding" by CATE type of heuristic approach, where the RBF matrix of features is binarized at the 90th percentile [42]. Next, the `RBF Cross Fitted` variant applies spectral clustering on RBF kernels of raw features instead of the cross fitted forest kernel. Then, the `RBF Oracle` method uses ground-truth CATE values to compute the idealized cluster-level statistics on the RBF of features to demonstrate the importance of clustering on the learnt kernel instead. Finally, we also evaluate the `CRE` package by [19], a recursive partitioning based method.

We evaluate methods using three metrics: The **Precision in Estimation of Heterogeneous Effects (PEHE)**, **within-cluster variance** ($V_{within}$), and the **between-cluster variance** ($V_{out}$). The PEHE quantifies the root mean squared error between predicted and true CATEs, aggregated across clustered estimates:

$$\text{PEHE} = \sqrt{\frac{1}{n} \sum_{i=1}^{n} \left( \hat{\tau}_i^{(\text{cluster})} - \tau_i \right)^2} \qquad (17)$$

where $\hat{\tau}_i^{(\text{cluster})}$ denotes the mean predicted treatment effect within the cluster to which unit $i$ belongs, and $\tau_i$ is the true individual treatment effect. The $V_{within}$ quantifies the heterogeneity within a cluster, while the $V_{out}$ quantifies the difference between clusters.

Results across cluster sizes ($k = 2$ to $6$) consistently highlight the effectiveness of our proposed clustering approach. Notably, substituting the feature space with the learned forest kernel substantially improves performance: `Cross Fitted` outperforms `RBF Cross Fitted` across all metrics, confirming the utility of CATE-informed similarity for clustering. While the `RBF Oracle` method—using ground truth treatment effects—yields the best performance among RBF-based baselines, it is still surpassed by our method in both estimation accuracy and cluster separability, further underscoring the strength of the forest-derived kernel even without access to the ground truth. We find that `CRE` performs competitively in PEHE and $V_{within}$, but shows low $V_{out}$—indicating similar predictions across clusters. While informative, our ablation setup (which fixes the number of clusters) is not the intended use case for `CRE`, which is designed to select the number of rules automatically via hyperparameters. Since the number of final rules cannot be specified directly, we relied on grid search. We also note that, unlike `grf`, the current `CRE` implementation cannot handle missing values, which may have affected its performance.

Finally, the thresholded binarization heuristic (`Thresholded`), while outperformed by our procedure, demonstrates competitive performance against the oracle despite its simplicity, and it can be viewed as a method to strengthen traditional feature based kernel clustering.

### C. Cluster Size Selection and Recovery Simulation

In this sub-section, we empirically test various cluster size selection methods for our method on a synthetic dataset.

We simulate a synthetic dataset to evaluate cluster recovery and treatment effect estimation. A total of $n = 1200$ samples are generated in two dimensions using the `mlbench.2dnormals` function, with $K \in \{2, \ldots, 6\}$ distinct clusters drawn from Gaussian distributions with standard deviation 0.6. The resulting features $X = (X_1, X_2) \in \mathbb{R}^2$ are standardized to have zero mean and unit variance. The baseline outcome function is defined as $\mu(x) = X_1$, and the potential outcomes are given by $Y(0) = \mu(x) + \epsilon$ and $Y(1) = Y(0) + \tau(x)$, where $\epsilon \sim \mathcal{N}(\mu = 0, \sigma = 0.5)$. Propensity scores are assigned using a logistic function of the form $e(x) = 1/\{1 + \exp(-[1.5X_2 - 0.5X_1])\}$, and treatment is assigned by sampling $T \sim \text{Bernoulli}(e(x))$.

For each run above, we apply our framework to first estimate, then cluster on the learnt kernel. We try out 4 popular methods for cluster size selection: The `eigengap` [43], `elbow` [44], `silhouette` [45], and the gap statistic (`gap_stat`) [46]. We evaluate the quality of the cluster recovery results via the Adjusted Rand Index (ARI) and the Normalized Mutual Information (NMI) (see Appendix A for definition). We present the results of the cluster recovery simulation design in Table II and display one example of a cluster recovery result in Figure 1 for illustration purposes.

| Method | ARI ↑ | NMI ↑ | PEHE ↓ |
|---|---|---|---|
| **eigengap** | **0.8873** | **0.8688** | **0.4989** |
| elbow | 0.7052 | 0.7128 | 0.8301 |
| gap_stat | 0.8209 | 0.7792 | 0.4993 |
| silhouette | 0.8510 | 0.8113 | 0.4993 |
| causal_forest | - | - | 0.5272 |

TABLE II: Average ARI, NMI, and PEHE for each cluster size selection method. PEHE of first step `causal_forest` included for comparison.

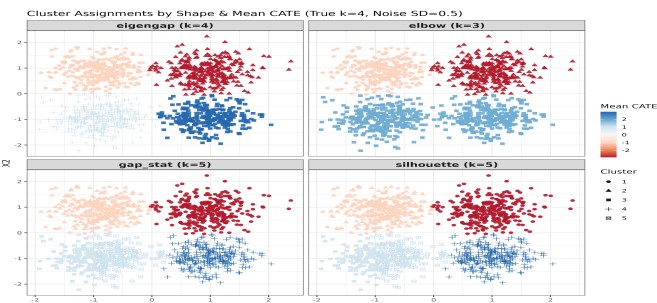

Fig. 1: Snapshot of one run of cluster recovery experiment with different cluster size selection methods

As shown in Table II, we find that the `eigengap` method consistently recovers the correct number of clusters and achieves the highest average performance across the cluster quality metrics. In addition, the bias-variance tradeoff of clustering is evident here, as the PEHE of the estimated CATE can actually be reduced by the clustering step due to increased within-cluster sample sizes.

### D. Adversarial Simulation Design

For this simulation design, we evaluate our framework as a plug-in ad-hoc CATE estimator on an adversarial structure with no true underlying distinct cluster subpopulations. We follow the simulation frameworks proposed by [5],[9], and [6] for evaluating our framework. We generate $n$ samples with $p$ features by drawing $X \sim U([0,1]^p)$. We fix the treatment assignment propensity as $e(X) = 0.5$ for all $X$, and set the baseline effect of the response to zero across features, i.e., $\forall i : Y_i^{(0)} = 0$. To assess robustness, we vary the sample size for training data in increments of 200, so that $n \in \{800, 1000, 1200\}$, while maintaining a fixed feature dimension $p = 20$. Additionally, we introduce iid Gaussian noise $\varepsilon_i \sim \mathcal{N}(0, \sigma)$ to each observed outcome $Y_i^{obs}$, with the standard deviation of noise $\sigma \in \{1, 2, 3, 4\}$. Model evaluation is conducted on a consistent unseen set of 2000 test samples (where only features are observed). We evaluate the excess risk of our clustering methodology against the underlying learner using the PEHE and Excess Risk metric:

$$\text{Excess Risk: } R(\hat{\tau}, \tau^*) := PEHE(\hat{\tau}) - PEHE(\tau^*), \quad (18)$$

where the excess risk defined in Equation (18) can be viewed as the performance penalty taken by our clustered estimate $\hat{\tau}$ over the underlying learner $\tau^*$. The ground truth CATE, $\tau(X)$, are generated based on the scenario defined in Equation (19):

$$\zeta(X) = 1 + \frac{2}{1 + \exp(-20(x - 1/3))} \quad (19)$$

$$\tau(X_i) = \zeta(X_{i1})\zeta(X_{i2}) \quad (20)$$

$$Y_i^{obs} = W_i * \tau(X_i) + \varepsilon_i \quad (21)$$

Following [5] and [6], we adopt this challenging response function due to its smooth, highly non-linear nature and the presence of numerous noisy irrelevant covariates and exogenous noise, in addition to the fact that no finite number of hard clusters can fully capture the true CATE without loss i.e., hard discrete assignment constraints handicap us versus continuous/soft assignments. A visualization of its non-trivial smooth structure is presented in Figure 3. We present our simulation results for the excess risk incurred by the clustering step in Figure 2. As expected, the greatest relative risk is incurred when noise is relatively low ($\sigma = 2$) scenario but rapidly decreases with a small increase in number of clusters. However, in high noise settings, this performance gap rapidly diminishes, enabling our approach to effectively capture challenging clusters with minimal reduction in estimation accuracy. Furthermore, as shown in Figure 4, even a relatively small number of clusters ($k = 6$) allows our method to reconstruct the level sets of the true response surfaces with limited fidelity loss.

### E. Observational Dataset Analysis: Synthea Emergency Health Records of Viral Sinusitis Treatment

For this section, we conduct causal clustering analysis based off the Synthea dataset [37], of patient emergency

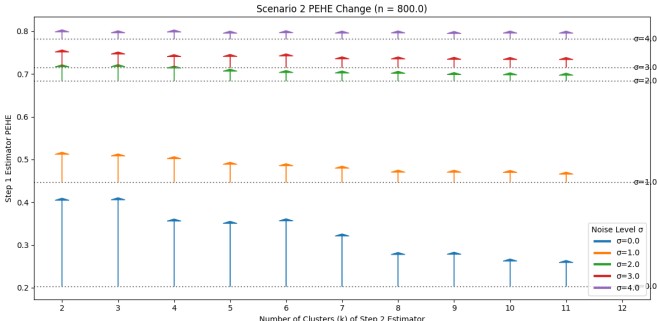

Fig. 2: Plot of excess risk incurred by clustering step against STEP 1 estimator's PEHE, with the simulated ground truth CATE defined in Equation (19). Each horizontal line indicates the original CATE estimator's PEHE at noise level $\sigma$. The size of the arrows indicates the performance penalty of the clustering step at that many clusters and noise level.

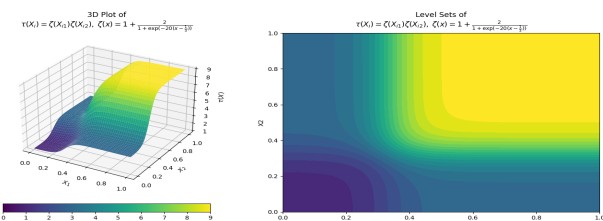

Fig. 3: Ground truth plots of CATE defined in Equation (19)

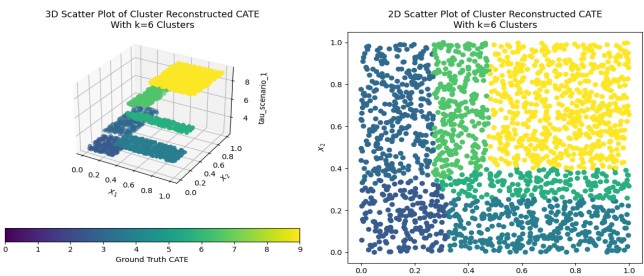

Fig. 4: Reconstructed CATE surface with cluster estimates of with $k = 6$ on the test dataset of $n = 2000, \sigma = 2$, trained on a separate $n = 1200, \sigma = 2$ samples drawn from the same distribution defined in Equation (19)

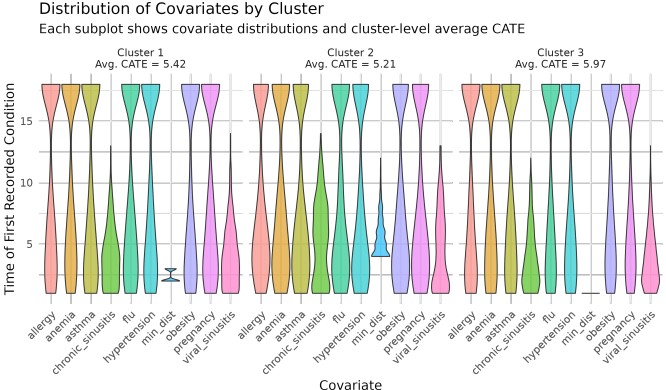

Fig. 5: Violin plots comparing covariate distributions across clusters derived from causal forest kernels. Each subplot corresponds to a single cluster.

health records (EHR) and outcomes based off their real life counterparts.

We utilize a cohort of 6,000 patients. Each patient is represented by a longitudinal record of emergency department visits, formatted as a structured table containing the patient's unique ID, the date of visit, and a list of diagnosis codes. All patients have been diagnosed with viral sinusitis at least once, which serves as the shared anchor condition.

This dataset records the outcomes for treatments of viral sinusitis. However, the treatment is confounded by the presence of a similar condition known as chronic sinusitis, which not only affects the propensity of treatment assignment, but also the efficacy of the treatment for the former: The treatment is slightly more effective if these 2 conditions occurred close in time to each other, but the propensity is inversely proportional to the difference in their recorded times, which attenuates the observed effectiveness if propensity is not adjusted for.

Following the setup and pre-processing of [38], the dataset includes timestamps of visits for a curated list of 7 other pre-treatment comorbidities: `asthma`, `allergy`, `anemia`, `flu`, `hypertension`, `obesity`, `pregnancy` to serve as irrelevant noise features. An additional continuous feature is derived for each patient: the average minimum distance between the two conditions. We then ran our framework and chose a cluster size of $k = 3$ via the `eigengap` method. We present the violin plots of features per cluster in Figure 5.

As shown in Figure 5, the 7 other irrelevant features display

the same distribution across clusters, while the relevant features: the first time visits for chronic sinusitis and viral sinusitis are the primary drivers of between cluster heterogeneity. This indicates that the clustering approach, guided by a learned causal kernel, prioritizes heterogeneity along meaningful and relevant dimensions, rather than arbitrary covariate differences.

## V. DISCUSSION AND CONCLUSION

We presented a novel framework that integrates kernelized clustering with debiased CATE estimation to uncover latent subpopulations that exhibit distinct responses to treatment. By leveraging the Robinson decomposition to orthogonalize the estimation problem and using the resulting learned kernel to guide clustering, our approach captures sample-level similarities rooted in treatment effect heterogeneity. This method not only reveals meaningful structure in complex, non-linear causal settings but also embeds a regularization perspective through residual-on-residual regression. Our pipeline is intuitive, modular, and compatible with widely used machine learning libraries. Experiments on semi-synthetic benchmarks and real-world data confirm the efficacy of our method. The framework thus offers a scalable and interpretable exploratory tool for causal subgroup discovery.

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

### APPENDIX A: CLUSTERING METRICS DEFINITIONS

In this section, we give the detailed formulae for the various clustering metrics used to evaluate our experiments.

*a) within-cluster variance ($V_{within}$):* The **within-cluster variance** $V_{\text{within}}$ measures the average deviation of predicted CATEs from their respective cluster mean:

$$V_{\text{within}} = \frac{1}{n} \sum_{c=1}^{k} \sum_{i \in \mathcal{C}_c} (\hat{\tau}_i - \bar{\tau}_c)^2 \tag{22}$$

where $\mathcal{C}_c$ is the set of units in cluster $c$, and $\bar{\tau}_c$ is the average predicted treatment effect in cluster $c$.

*b) between-cluster variance ($V_{out}$):* The **between-cluster variance** $V_{\text{out}}$ quantifies the dispersion of cluster-level average treatment effects around the global mean:

$$V_{\text{out}} = \frac{1}{n} \sum_{c=1}^{k} |\mathcal{C}_c| \cdot (\bar{\tau}_c - \bar{\tau})^2 \tag{23}$$

where $\bar{\tau}$ is the overall mean of all predicted treatment effects.

*c) Rand Index (RI).:* The Rand Index is a classical measure of similarity between two clusterings. It considers all unordered pairs of samples and counts the proportion of pairs whose cluster memberships are either the same in both clusterings or different in both clusterings. Formally, let $a$ denote the number of pairs of samples that are in the same cluster in both the predicted and ground-truth clusterings, and let $d$ denote the number of pairs that are in different clusters in both clusterings. Then the Rand Index is defined as:

$$\text{RI} = \frac{a + d}{\binom{n}{2}}, \tag{24}$$

where $\binom{n}{2}$ is the total number of unique sample pairs. The RI takes values between $0$ and $1$, with $1$ indicating perfect agreement and $0$ indicating complete disagreement.

*d) Adjusted Rand Index (ARI).:* The Adjusted Rand Index is a measure of agreement between the predicted cluster labels and the ground-truth labels. It adjusts the classical Rand Index for chance agreement:

$$\text{ARI} = \frac{\text{RI} - \mathbb{E}[\text{RI}]}{\max(\text{RI}) - \mathbb{E}[\text{RI}]} \tag{25}$$

where RI denotes the Rand Index, which counts the number of pairwise agreements in cluster membership between the predicted and true clusterings. ARI ranges from $-1$ (complete disagreement) to $1$ (perfect agreement), with $0$ indicating random labeling.

*e) Normalized Mutual Information (NMI).:* The Normalized Mutual Information quantifies the amount of information shared between the predicted and true clusterings. Let $U$ and $V$ denote the sets of predicted and true clusters respectively, and let $I(U;V)$ be their mutual information:

$$\text{NMI}(U, V) = \frac{I(U;V)}{\sqrt{H(U)H(V)}} \tag{26}$$

where $H(U)$ and $H(V)$ are the entropies of the predicted and true clusterings. NMI ranges from $0$ (no mutual information) to $1$ (perfectly matched clusterings), and is symmetric.

### APPENDIX B: BEYOND HEALTHCARE

We analyze the well-known LaLonde Temporary Employment Program dataset [47], which tracks the income effects of a randomized employment intervention on 614 individuals. The original study highlighted the limitations of non-experimental methods due to the substantial divergence in Average Treatment Effect (ATE) estimates and subsequent research [48],[49] were instrumental in demonstrating the importance of propensity score adjustments and covariate balance based methods.

The dataset includes a binary treatment indicator for program assignment and records earnings in 1978 as the outcome. Covariates include years of education, race, age, and pre-treatment earnings from 1974 and 1975.

Figure 6 visualizes covariate distributions across clusters obtained using a causal forest-derived similarity kernel. Each subplot corresponds to a single cluster and displays violin plots for all covariates within that group. This view highlights how individual clusters are internally composed in terms of feature distribution. In contrast, Figure 7 reorganizes the perspective: each subplot corresponds to a single feature and shows its distribution across all clusters, making it easier to identify which covariates drive between-cluster separation.

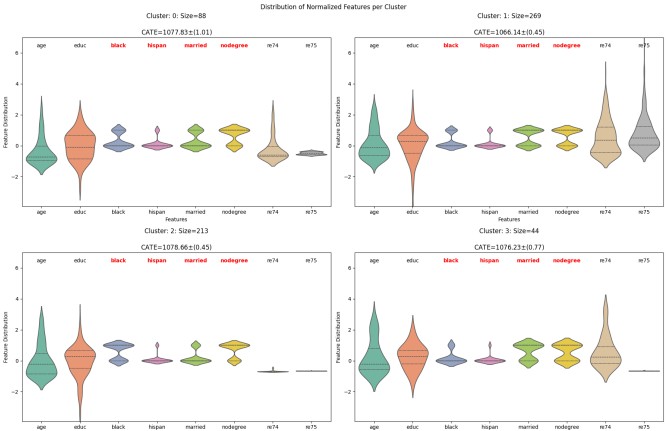

Fig. 6: Cluster-level violin plots of covariate distributions using a causal forest-derived kernel. Each subplot corresponds to a cluster, with violin plots for all features within that cluster. Binary features are highlighted in red. Cluster-specific CATE means and standard errors are reported in the subplot titles.

Our results yield several noteworthy findings. First, the cluster-specific bootstrapped CATE estimates in Figure 6 are consistent with the established ATE estimate of 1079.13 obtained via propensity score matching [50], lending support to the validity of our subgrouping approach. Second, the covariates that most strongly differentiate clusters (`re74` and `re75`) correspond to pre-treatment earnings, which are well-known

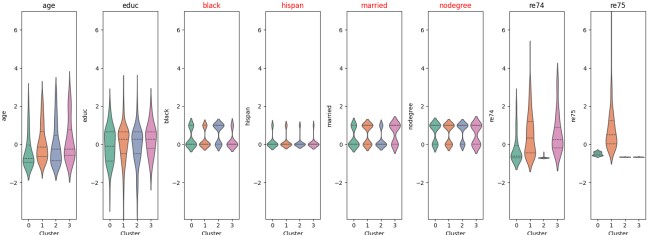

Fig. 7: Feature-level violin plots comparing covariate distributions across clusters derived from causal forest kernels. Each subplot corresponds to a single feature. Binary features are highlighted in red.

confounders in non-experimental variants of the LaLonde dataset [49]. This indicates that the clustering approach, guided by a learned causal kernel, prioritizes heterogeneity along substantively meaningful and causally relevant dimensions, rather than arbitrary covariate differences. These findings suggest that our method can generalize to settings outside of healthcare and remains interpretable and useful in subgroup discovery.