# OpenReview forum: "Causal Clustering for Conditional Average Treatment Effects Estimation and Subgroup Discovery"
_IEEE.org/EMBS/BHI/2025/Conference — BHI 2025_

### Official Review · Reviewer_mL3j · 2025-07-03
**The experiments look good except for baseline.**

**Confidence:** 3
**Clarity Of Writing:** good
**Clinical Significance:** good
**Methodological Novelty:** good
**Overall Rating:** 6

**Experiments And Results:**

fair

**Questions For The Authors:**

How does the hyper-parameter (λ, k) affect? Will they be sensitive to the results?

**Strengths:**

1. Methodological novelty - integrates kernel learning from causal forests with convex/spectral clustering.
2. Empirical support - synthetic and real-world data are all experimented.
3. Interpretability: hard clusters give clear subgroup-level CATEs, aiding policy or clinical decisions.

**Summary Of The Paper:**

The paper proposes a two-stage framework that first estimates debiased CATEs with a kernel matrix and then clusters individuals in that learned similarity space. The kernelized clustering within the residual-on-residual regression framework is presented and exhaustive experiments were conducted to show the effectiveness of the method.

**Weaknesses:**

1. Limited comparisons – omits recent transformer or tree-based partitioning baselines.
2. What about other models' performance on that?

---

### Official Review · Reviewer_nT24 · 2025-07-15
**review for Causal Clustering for conditional average treatment effects estimation and subgroup discovery**

**Confidence:** 3
**Clarity Of Writing:** fair
**Clinical Significance:** great
**Methodological Novelty:** good
**Overall Rating:** 4

**Experiments And Results:**

poor

**Questions For The Authors:**

1.	Two real data examples used in this paper are based on RCT studies, it is not clear whether the proposed method can be used for observational data. If so, how?

2.	Figure 1, it is not clear to me how subjects in the top panel be grouped into three subpopulations with negative, none and positive response. It seems to be very arbitrary.

**Strengths:**

The paper introduces an innovative approach by combining causal inference with kernelized clustering, using a learned kernel derived from causal forests. This method leverages the Robinson decomposition and R-Learner framework to estimate debiased CATEs, followed by clustering to uncover latent subgroups with heterogeneous treatment effects. The proposed framework is designed to be compatible with standard machine learning libraries, making it accessible and reproducible. Its modularity allows for integration with existing model selection and tuning pipelines, enhancing its practical applicability for researchers and practitioners.

**Summary Of The Paper:**

The paper introduces an approach to identify subpopulations with distinct treatment effect by combining causal inference with kernelized clustering, using a learned kernel derived from causal forests.

**Weaknesses:**

1.	In general, clustering based techniques are very exploratory with several limitations. First, it is challenging to demonstrate the validity of the identify subpopulations. Second, it does not provide information directly regarding why the subpopulations have different treatment effect and as a result, provide limited clinical utilities in guiding personalized treatment.
2.	In the paper, the authors claim that “CATE-based clustering, on the contrary, aims to recover patient groupings that are causally meaningful and clinically actionable”. Due to the limitation of clustering techniques discussed above, I cannot agree that patient subgroups identified by clustering techniques are clinically actionable.
3.	At the bottom of page 2, authors reviewed several related works that are based on the two-stage approach but failed to discuss the advantage of the new method proposed in paper as compared with existing methods.
4.	Most importantly, I don’t see the advantage of the method proposed in paper, or clustering based approach in general, as compared with method like causal forest (S Athey, 2016).
5.	It is not clear how to number of clusters will be determined which is the most important decision when conducting clustering analysis. It is not clear to me why the authors designed the simulation study with no true underlying distinct cluster subpopulations. I think many readers, including myself, would be very interested in knowing how the proposed method be able to identify the right number of clusters.

---

### Official Review · Reviewer_hiaN · 2025-07-15
**Causal Clustering for Conditional Average Treatment Effects Estimation and Subgroup Discovery**

**Confidence:** 3
**Clarity Of Writing:** great
**Clinical Significance:** good
**Methodological Novelty:** great
**Overall Rating:** 7

**Experiments And Results:**

excellent

**Questions For The Authors:**

How sensitive is the method to the quality of the base CATE estimator?
How should practitioners choose the number of clusters (k) in the absence of ground truth subgroup labels?
Can you include real-world healthcare examples where subgroup heterogeneity is important?

**Strengths:**

1. The kernel-based clustering with causal effect estimation approach is innovative and interpretable in subgroup discovery.
2. Provides theoretical justification for causal inference computation.
3. comprehensive evaluation including ablation studies, synthetic benchmarks, and real-world examples,

**Summary Of The Paper:**

This paper introduces a novel framework that integrates kernel-based clustering with CATE (Conditional Average Treatment Effect) estimation for subgroup discovery in causal inference.

**Weaknesses:**

1. performance is dependent of initial estimation of CATE.
2. Choice of λ (regularization) and number of clusters (k) can influence results;
3. real world data showed are not related to healthcare.